# Current Practices and Opportunities for Outpatient Parenteral Antimicrobial Therapy in Hospitals: A National Cross-Sectional Survey

**DOI:** 10.3390/antibiotics11101343

**Published:** 2022-10-01

**Authors:** Hester H. Stoorvogel, Marlies E. J. L. Hulscher, Heiman F. L. Wertheim, Ed P. F. Yzerman, Maarten Scholing, Jeroen A. Schouten, Jaap ten Oever

**Affiliations:** 1Radboud University Medical Center, Radboud Institute for Health Sciences, Department of Internal Medicine & IQ Healthcare, Radboud Center for Infectious Diseases, 6500HB Nijmegen, The Netherlands; 2Radboud University Medical Center, Department of IQ Healthcare, Radboud Center for Infectious Diseases, 6500HB Nijmegen, The Netherlands; 3Radboud University Medical Center, Department of Medical Microbiology, Radboud Center for Infectious Diseases, 6500HB Nijmegen, The Netherlands; 4ABR Zorgnetwerk Noord-Holland–Flevoland, 1105AZ Amsterdam, The Netherlands; 5OLVG Lab BV, 1091AC Amsterdam, The Netherlands; 6Radboud University Medical Center, Department of Intensive Care & IQ Healthcare, Radboud Center for Infectious Diseases, 6500HB Nijmegen, The Netherlands; 7Radboud University Medical Center, Department of Internal Medicine, Radboud Center for Infectious Diseases, 6500HB Nijmegen, The Netherlands

**Keywords:** outpatient parenteral antimicrobial therapy (OPAT), antimicrobial stewardship, antimicrobial resistance, home infusion therapy, survey

## Abstract

This nationwide study assessed how outpatient parenteral antimicrobial therapy (OPAT) is organised by Dutch acute care hospitals, the barriers experienced, and how an OPAT program affects the way hospitals organised OPAT care. We systematically developed and administered a survey to all 71 Dutch acute care hospitals between November 2021 and February 2022. Analyses were primarily descriptive and included a comparison between hospitals with and without an OPAT program. Sixty of the 71 hospitals (84.5%) responded. Fifty-five (91.7%) performed OPAT, with a median number of 20.8 (interquartile range [IQR] 10.3–29.7) patients per 100 hospital beds per year. Of these 55 hospitals, 31 (56.4%) had selection criteria for OPAT and 34 (61.8%) had a protocol for laboratory follow-up. Sixteen hospitals (29.1%) offered self-administered OPAT (S-OPAT), with a median percentage of 5.0% of patients (IQR: 2.3%–10.0%) actually performing self-administration. Twenty-five hospitals (45.5%) had an OPAT-related outcome registration. The presence of an OPAT program (22 hospitals, 40.0%) was significantly associated with aspects of well-organised OPAT care. The most commonly experienced barriers to OPAT implementation were a lack of financial, administrative, and IT support and insufficient time of healthcare staff. Concluding, hospital-initiated OPAT is widely available in the Netherlands, but various aspects of well-organised OPAT care can be improved. Implementation of a team-based OPAT program can contribute to such improvements. The observed variation provides leads for further scientific research, guidelines, and practical implementation programs.

## 1. Introduction

Outpatient parenteral antimicrobial therapy (OPAT) is increasingly provided for a wide variety of infections [1]. It is a safe and cost-effective way to shorten or even avoid hospital admissions [2,3]. Continuation of intravenous therapy in the home situation increases patients’ freedom and autonomy [4,5], which can be further enhanced if patients administer the antimicrobials themselves [6]. Consequently, patient satisfaction with OPAT is high [2,7]. 

Due to these benefits, OPAT contributes to the goals of antimicrobial stewardship, namely, to ensure the appropriate use of antimicrobials in the broadest sense of the word [8]. This is one of the reasons why it is thought that OPAT should be integrated into antimicrobial stewardship (AMS) programs [9]. Like antimicrobial stewardship, the quality of OPAT care benefits from a well-organised approach. A care bundle approach that includes interventions throughout the OPAT care pathway with, for example, consultation with an infectious disease (ID) physician or clinical microbiologist, patient screening and education and monitoring leads to fewer readmissions [10]. Additionally, organising OPAT care in such a way that the treating physician has direct access to laboratory results is associated with fewer readmissions [11].

Recent international clinical practice guidelines provide clinicians with guidance for implementing such an organised OPAT service [9,12]. Furthermore, recently developed quality indicators (QIs) that define well-organised OPAT care can support the implementation of an OPAT program and help measure the quality of OPAT care—a prerequisite for improving quality of care [13].

Unfortunately, however, studies repeatedly show that practice—including appropriate antibiotic use—deviates from recommended care [14,15]. Deviations and practice variation can indicate knowledge gaps, but also other barriers such as organisational and financial constraints [16,17]. Knowledge about these aspects can guide the focus of scientific research and provide other opportunities for intervention. 

Therefore, the primary aim of this study was to assess the extent to which OPAT is provided by Dutch acute care hospitals, how they organise OPAT care, and what barriers they experience to implement OPAT. The secondary aim of the study was to determine the association between the presence of an OPAT program, with the presence of an OPAT team as the defining characteristic, and how OPAT care was organised. 

## 2. Materials and Methods

### 2.1. Study Design, Setting, and Participants

We conducted a cross-sectional survey among all 71 acute care hospitals in the Netherlands between November 2021 and February 2022. 

The Dutch organisation of hospital care and antimicrobial stewardship teams are described in the web-only Appendix A [18,19,20]. In the Netherlands, OPAT is coordinated from the hospitals. It is always provided at a patient’s home or in a nursing home; there are no infusion centres. The antimicrobial therapy can both be administered by home care nurses or by patients and/or their caregivers themselves. Either the hospital pharmacy or an external pharmaceutical company prepares and delivers the antimicrobials. Similar to inpatient antimicrobial treatment, almost all antimicrobials and OPAT care, including the home care, are reimbursed by Dutch health insurance companies. 

### 2.2. Survey Development and Administration

We systematically developed an electronic survey, using four steps to ensure an evidence-based research tool, see web-only Appendix A [9,12,13,16,20,21,22,23,24,25,26,27,28,29]. An ID physician and an epidemiologist who were not involved in the development process provided feedback on the survey. Next, a pilot was performed in one hospital to identify final issues and estimate the time needed to complete the survey. The survey covered the following topics: hospital characteristics, OPAT performance and OPAT program/team, OPAT practices, monitoring of OPAT-related outcome measures, and barriers to OPAT care. The electronic survey was built in LimeSurvey (LimeSurvey, Hamburg, Germany).

The first author (HS) sent the survey via a link in an email to the antimicrobial stewardship team’s contact person of the Dutch working party on Antibiotic Policy (SWAB). The SWAB aims to support antimicrobial stewardship teams and, among other things, reports annually on the composition and activities of antimicrobial stewardship teams, with this year’s focus on OPAT [19]. Potential participants were asked to forward the survey to a colleague if that person was more knowledgeable about the topics in the survey. If applicable, the contact person received up to three reminders by e-mail or telephone. No incentives for participation were provided. 

### 2.3. Data Analysis

We included only hospitals that had completed all survey questions. The results are primarily descriptive and are presented using the median (with interquartile range [IQR], 25th–75th percentile) and absolute numbers/proportions. In addition, the Mann–Whitney U test and chi square tests were used to compare continuous outcomes and proportions, respectively, between hospitals with and without an OPAT program. An OPAT program was defined as the presence of a team in the hospital whose roles and responsibilities with respect to OPAT, as well as the OPAT care pathway, were defined. In the analysis of barriers to OPAT care, the answer ‘Do not know’ was considered a missing value, leading to different denominators. Analyses were performed with SPSS Statistics 25 (IBM, Armonk, NY, USA).

### 2.4. Ethical Considerations

According to Dutch legislation, this study did not require review by a research ethics committee. Prior to the start of the survey, participants received information about the study and were asked for informed consent.

## 3. Results

### 3.1. Response and Hospital Characteristics

Sixty of the 71 Dutch acute care hospitals completed the survey (84.5%). Of the 11 missing hospitals, four hospitals submitted partial responses that we excluded and the other seven hospitals did not submit any response. All seven university hospitals responded (100.0%), 40 of the 44 non-university teaching hospitals (90.9%), and 13 of the 20 non-teaching hospitals (65.0%). One of the responding hospitals was a specialised paediatric hospital. The 60 hospitals had a median of 400 hospital beds (IQR: 298–652), compared to a median of 350 hospital beds (IQR: 200–600) of the 11 non-responding hospitals. 

### 3.2. OPAT Performance and OPAT Program 

Fifty-five of the 60 responding hospitals performed OPAT (91.7%), including the paediatric hospital. Of the 55 OPAT-performing hospitals, 22 (40.0%) had a team-based OPAT program. The number of hospital beds was similar between hospitals with and without a program (with: 440.5 [IQR 291.5–681.3] vs. without 410.0 [IQR 277.5–663.5], *p* = 0.952), as was the distribution of hospital type (university hospital 18.2% vs. 9.1%, non-university teaching hospital 63.6% vs. 66.7%, and general non-teaching hospital 18.2% vs. 24.2%, *p* = 0.578). The five hospitals (8.3%) that indicated they did not perform OPAT had a median of 318 hospital beds (IQR: 210–600). Four of these hospitals were non-university teaching hospitals and one was a general non-teaching hospital. The barriers these hospitals experienced to perform OPAT were diverse and are described in web-only Appendix A.

The characteristics of the 22 OPAT teams are shown in Table 1. Seventeen OPAT teams (77.3%) fell under the responsibility of the hospital’s antimicrobial stewardship team. Overall, the hospitals indicated that the OPAT team takes over the medical responsibility from the treating physician of only 7.5% (IQR: 0.0–42.5) of the patients. Only two hospitals indicated that their OPAT teams became medically responsible for all OPAT patients. 

### 3.3. Current OPAT Practices in The Netherlands

The median number of OPAT patients in the 55 hospitals that performed OPAT was 20.8 per 100 hospital beds per year, with substantial variation between hospitals as indicated by a broad IQR (10.3–29.7), see also web-only Appendix A. Patient volume was not associated with the presence of a program (with: 21.5 [IQR 11.0–33.1] vs. without 20.8 [IQR 10.0–29.6], *p* = 0.414). 

The current practices of OPAT care in the Netherlands are shown in Table 2. All hospitals initiated OPAT in hospitalised patients, administering the first dose before discharge. However, no hospital administered the first dose (i.e., initiated OPAT) at the emergency department if hospital admission was not necessary. The presence of an OPAT program was significantly associated with an increased likelihood of initiating OPAT at the day care facility for outpatients (59.1% vs. 18.2%; *p* = 0.002). Inclusion and exclusion criteria for the eligibility for OPAT and a protocol for laboratory follow-up during OPAT were present in over half of the responding hospitals. Hospitals with an OPAT program were more likely to have inclusion and exclusion criteria (86.4% vs. 36.4%; *p* = 0.000) and a protocol for laboratory follow-up (86.4% vs. 45.5%; *p* = 0.002). A consultation by an ID specialist was mandatory in only 12 hospitals (21.8%) before the start of OPAT; this was numerically more frequent in hospitals with an OPAT program (31.8% vs. 15.2%; *p* = 0.143). In both hospitals with and without an OPAT program, approximately two thirds reported the start of OPAT to the patient’s general practitioner or geriatrist. Sixteen hospitals (29.1%) indicated that the possibility of self-administration of OPAT (S-OPAT) is offered in their hospital, corresponding to a median percentage of 5.0% of patients performing S-OPAT (IQR: 2.3–10.0%). 

### 3.4. Monitoring of OPAT-Related Outcome Measures

More than half of the hospitals (54.5%) reported no outcomes being measured for OPAT patients. Of the 25 hospitals (45.5%) that measured outcomes, completion of OPAT as planned (64.0%), clinical outcome (64.0%), and side effects (60.0%) were the most commonly measured metrics, whereas patient survival (20.0%) and patient experiences (20.0%) were the least frequently monitored, see Figure 1. Seventy percent of hospitals without an OPAT program measured no outcomes at all, compared to only 30% of hospitals with a program (*p* = 0.006). Within the hospitals that measured outcomes, patient survival and readmission were significantly more often monitored by hospitals with an OPAT program than those without (Figure 1).

### 3.5. Experienced Barriers to OPAT Care

Figure 2 shows the extent to which potential barriers to OPAT care were experienced by the 55 OPAT performing hospitals. Lack of funding, time available for OPAT, IT support, and administrative support were most regularly perceived to hinder the performance of OPAT. Most hospitals disagreed that a lack of interest in OPAT among eligible patients was a barrier to OPAT, although it should be noted that 14 hospitals answered ‘Do not know’ to this question. The differences in barriers between hospitals with and without an OPAT program are presented in web-only Appendix A.

## 4. Discussion

The results of our systematically developed nationwide survey showed that most hospitals in the Netherlands have an OPAT service. However, there seemed to be an untapped patient potential, since the number of patients receiving OPAT annually differed between hospitals of similar size. Furthermore, only 40% of Dutch hospitals performing OPAT had a team-based OPAT program, whereas their presence was associated with several aspects of well-organised OPAT care. Certain aspects could be improved in the entire OPAT care pathway, such as the presence of criteria for the selection of patients, the inclusion of patients at the emergency department to avoid hospital admission, medical responsibility by the OPAT team, the provision and organisation of self-administration, and monitoring outcome measures. The most common barriers encountered in providing OPAT care were lack of funding, personnel, and IT and administrative support. 

A high percentage of Dutch hospitals (95%) performed OPAT compared to other countries (60–75%) [16,19,20]. Despite the wide availability, the variation observed in number of OPAT patients per 100 hospital beds annually suggests that more patients could benefit from OPAT. Paediatricians were not involved in the OPAT teams of the non-paediatric hospitals, suggesting that children may not have the opportunity to receive outpatient IV antimicrobial treatment [30]. In addition, no patients were discharged with OPAT from the ED to home to avoid hospital admission, whereas with proper selection this could be done safely, resulting in improved patient satisfaction [31]. Furthermore, only 30% of Dutch hospitals were found to offer S-OPAT, with an estimated five percent of all OPAT patients actually performing S-OPAT. Possibly caused by less organised homecare or greater distances, the proportion of S-OPAT patients is higher (15–50%) in the United Kingdom and the United States [16,20,23]. With the prerequisite that patients are properly trained, S-OPAT should be encouraged as it is considered safe, cost-effective, and contributes—in addition to OPAT itself—to increased person-centeredness [4,9,12].

Several aspects of well-organised OPAT care showed room for improvement compared to the recommendations of international guidelines, although some recommendations may be disputed. One was the presence of selection criteria for patients eligible for OPAT. Although the good practice recommendations of the British Society for Antimicrobial Chemotherapy (BSAC) mention general aspects of importance, they do not describe explicit criteria [9]. Besides, it is acknowledged that several personal factors influence the likelihood of OPAT failure or complications [9]. The OPAT guideline of the Infectious Diseases Society of America (IDSA) states that almost all patients are eligible for some form of OPAT, with the main challenge being to determine the appropriate setting for OPAT [12]. Together, this means that starting OPAT is a subjective consideration in which the risk of adverse events should be weighed individually, and where documented general principles would be of greater value to OPAT teams than rigid selection criteria.

Second, hospitals often did not have a local protocol for laboratory follow-up during OPAT. Structural monitoring of laboratory results has been associated with fewer readmissions [11]. However, antimicrobials differ in terms of toxicity [12], most adverse events presumably occur in the first two weeks [32], and it is likely that patient factors are influential. Consequently, the frequency of monitoring needs to be adjusted to individual factors, while evidence is gathered on what to monitor when in which patient [13]. 

Third, outcomes of OPAT care were not often measured, similar to other studies [9,23], despite the fact that monitoring is a prerequisite for improving care. With less than half of the hospitals measuring OPAT related outcomes, there is still much to be gained. 

A formal OPAT program was present in only 40% of the hospitals, whereas a program was associated with an increase in outpatient OPAT initiation, presence of selection criteria, laboratory follow-up, and measurement of outcomes. Literature has consequently shown that these aspects of care can indeed lead to higher quality of care and fewer readmissions [10,11,13]. Interestingly, Dutch OPAT teams on average did not often take over medical responsibility for patients indicated for OPAT. When the medical responsibility remains with the physician, the OPAT team primarily organises the logistics surrounding OPAT and is responsible for certain aspects of patient communication. Among other things, OPAT team members inform and instruct patients on OPAT, ensure good communication among all stakeholders involved, make sure that follow-up visits and laboratory tests take place, and can be contacted for certain problems after discharge. Fulfilling a merely facilitating role is not necessarily harmful if the responsible healthcare professional provides close follow-up [33], has the necessary expertise, and can consult a medical microbiologist or ID specialist if necessary. On the other hand, it is plausible that there are more patient subgroups that would benefit from the OPAT team taking over medical responsibility [34]. Similarly, for the performance of bedside consultation by an ID specialist before starting OPAT, it is likely that a balance exists between too much involvement (i.e., large time investment and high costs) and not using the appropriate expertise, whereby it must be said that a 20% mandatory bedside consultations is quite low [9,12,26].

Our study provides clinical and scientific implications that are generalisable to other countries as well. Our study is a clear case for the establishment of an OPAT team, as it was associated with well-organised OPAT care. Furthermore, some of the practice variation may be attributed to a lack of evidence, for example, regarding the frequency of laboratory monitoring and perhaps the safest, most efficient, and patient-centred method of follow-up. In addition, some recommendations could be less rigid (e.g., selection criteria and becoming medically responsible), with OPAT teams better served by practical advice on making individual decisions. Finally, OPAT teams, also outside the Netherlands [16], need better support at all levels. International or national documents recommending, for example, a staffing standard, could improve structural prerequisites by influencing policy makers and managers, as has already been done for an OPAT pharmacist to patient ratio and antimicrobial stewardship in general [18,35]. An exchange of practical information, for example, on the organisation of S-OPAT, may also be useful for OPAT teams, as educational and training materials are needed to start or expand new models of care.

The high response rate (84.5%) allowed us to get a nationwide overview of OPAT care in the Netherlands. Another strength was the systematic development of a survey encompassing the OPAT care pathway compared to previous OPAT surveys, see the publications in web-only Appendix A. Nevertheless, our study also has limitations. First, survey studies rely on the self-reporting of their participants and are thus subject to respondent bias and recall bias. To minimise this bias and ensure validity, we provided explanations for potentially misinterpreted terms in the survey, and we verified some unexpected answers. Second, OPAT is not provided in so-called infusion centres in the Netherlands, but only through home-based infusion and sporadically in skilled nursing facilities. Nevertheless, we believe that the aspects of care surveyed are also applicable to the other OPAT care delivery models. Third, we did not focus on specific patient groups, such as children or the frail elderly. This would be an interesting focus for further research.

In conclusion, our study demonstrates a wide availability of OPAT services in Dutch hospitals, although the extent to which and the way in which several aspects of OPAT care are performed could be improved. Our data indicate that a formal OPAT program can contribute to such improvements. Furthermore, efforts are needed to remove organisational barriers and improve support to OPAT teams.

## Figures and Tables

**Figure 1 antibiotics-11-01343-f001:**
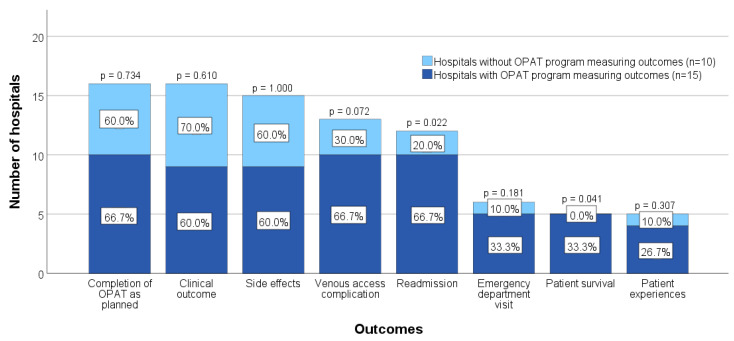
Comparison of outcomes measured by hospitals with and without an OPAT program. The percentages represent the proportion of hospitals with or without a program measuring the specific outcome, and the *p*-value is the significance of the comparison between these proportions.

**Figure 2 antibiotics-11-01343-f002:**
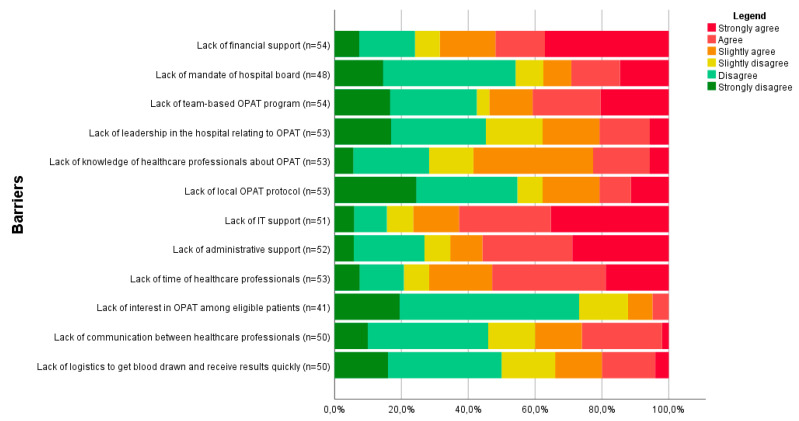
Barriers to OPAT care experienced by OPAT performing hospitals (*n* = 55) ^a^. ^a^ The value ‘Do not know’ was considered as missing, resulting in different denominators per barrier.

**Table 1 antibiotics-11-01343-t001:** Characteristics of the OPAT teams.

Characteristic	With OPAT Program ^a^ (*n* = 22)
** * Composition of OPAT ^b^ team * **	
Pharmacist	20 (90.9%)
Infectious disease physician	18 (81.8%)
Transfer of care nurse/employee ^c^	17 (77.3%)
Nurse specialist	11 (50.0%)
Clinical microbiologist	10 (45.5%)
Pharmacist assistant ^d^	5 (22.7%)
Nurse	5 (22.7%)
Home care nurse/employee	2 (9.1%)
Internal medicine specialist	1 (4.5%)
OPAT physician	1 (4.5%)
Paediatric oncologist	1 (4.5%)
Administrative assistant	1 (4.5%)
IT specialist ^e^	1 (4.5%)
Paediatrician	0 (0%)
**OPAT program falling under the responsibility of the AMS ^f^ team**	17 (77.3%)
**Annual frequency for policy meetings ^g^ of OPAT team, median (IQR ^h^)**	4.0 (1.8–6.0)
**Percentage of patients where the OPAT team takes on medical responsibility, median (IQR)**	7.5% (0.0–42.5)

^a^ Values are *n* (%) unless otherwise indicated. ^b^ Outpatient parenteral antimicrobial therapy. ^c^ An employee responsible for organising the care a patient needs after discharge, either at home or in a nursing home. ^d^ An employee who has a supporting role in the pharmacy and informs patients and assists in ordering, preparing, and distributing medication. ^e^ Information technology specialist. ^f^ Antimicrobial stewardship. ^g^ Meeting on functioning of the team, (future) policies, etc. ^h^ Interquartile range (25th–75th percentile).

**Table 2 antibiotics-11-01343-t002:** Organisation of OPAT care compared between hospitals with and without an OPAT program.

Characteristic	Total ^a^ (*n* = 55)	With OPAT Program ^a^ (*n* = 22)	Without OPAT Program ^a^ (*n* = 33)	*p*-Value ^b^
** * Location of first dose of antimicrobials ^c^ * **				
At home for outpatients	8 (14.5%)	3 (13.6%)	5 (15.2%)	0.876
At the day care facility for outpatients	19 (34.5%)	13 (59.1%)	6 (18.2%)	0.002
At the emergency department when hospital admission is unnecessary	0 (0%)	0 (0%)	0 (0%)	n.a.
In the patient ward before discharge	55 (100.0%)	22 (100.0%)	33 (100.0%)	n.a.
** * Intravenous access device (<7 days) most frequently used * **				0.794
Peripherally inserted central catheter (PICC)	26 (47.3%)	11 (50.0%)	15 (45.5%)	
Peripheral intravenous catheter	23 (41.8%)	8 (36.4%)	15 (45.5%)	
Central vascular access device (other than PICC)	2 (3.6%)	1 (4.5%)	1 (3.0%)	
Midline	1 (1.8%)	0 (0%)	1 (3.0%)	
Not applicable ^d^	3 (5.5%)	2 (9.1%)	1 (3.0%)	
** * Intravenous access device (>7 days) most frequently used * **				0.329
Peripherally inserted central catheter (PICC)	48 (87.3%)	18 (81.8%)	30 (90.9%)	
Peripheral intravenous catheter	0 (0%)	0 (0%)	0 (0%)	
Central vascular access device (other than PICC)	3 (5.5%)	1 (4.5%)	2 (6.1%)	
Midline	4 (7.3%)	3 (13.6%)	1 (3.0%)	
** * Infusion administration method most frequently used * **				0.246
Electronic infusion pump	31 (56.4%)	10 (45.5%)	21 (63.6%)	
Elastomeric device	24 (43.6%)	12 (54.5%)	12 (36.4%)	
**Presence of in- and exclusion criteria for OPAT**	31 (56.4%)	19 (86.4%)	12 (36.4%)	<0.001
**Presence of a protocol for laboratory follow-up**	34 (61.8%)	19 (86.4%)	15 (45.5%)	0.002
**Consultation with infectious disease physician mandatory before start OPAT**	12 (21.8%)	7 (31.8%)	5 (15.2%)	0.143
**Start of OPAT reported to general physician or geriatrist**	37 (67.3%)	15 (68.2%)	22 (66.7%)	0.907
**Possibility of self-administered OPAT by patient or carer**	16 (29.1%)	8 (36.4%)	8 (24.2%)	0.332
**Presence of in- and exclusion criteria for S-OPAT ^e^**	4 (25.0%)	2 (25.0%)	2 (25.0%)	1.000
**Presence of a training program for S-OPAT**	3 (18.8%)	0 (0%)	3 (37.5%)	0.055
**Percentage of patients performing S-OPAT, median (IQR)**	5.0% (2.3–10.0)	5.0% (4.0–17.8)	3.5% (1.0–10.0)	0.442

^a^ Values are *n* (%) unless otherwise indicated; ^b^ Significance of the differences between hospitals with and without an OPAT program; ^c^ Multiple answers possible; ^d^ Not applicable: Three hospitals indicated that they did not perform OPAT for less than seven days; ^e^ Self-administered outpatient parenteral antimicrobial therapy.

## Data Availability

The data that support the findings of this study are available from the corresponding author upon reasonable request.

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
