# Peer review of "Current Practices and Opportunities for Outpatient Parenteral Antimicrobial Therapy in Hospitals: A National Cross-Sectional Survey"

_antibiotics, 2022, doi:10.3390/antibiotics11101343_

Round 1
Reviewer 1 Report
The authors have put together a well written paper that provides an overview of OPAT services in Netherlands.
A few comments/suggestions are listed below:
1) Language: kindly stick to either American or British English (e.g. paediatrician vs pediatrician)
2) Tables: Kindly provide the meaning of abbreviations in the footnotes - e.g. in Table 1, provide meaning of AMS, IQR, OPAT
3) Discussion:
a) Second paragraph: the authors state that: 'No paediatricians were involved in the OPAT teams .... '. Could the authors clarify whether Children's Hospitals participated in the study and what was their responses to the questionnaires?
b) Third paragraph: Kindly provide the meaning of BSAC
Reviewer 2 Report
This is a survey of all Danish hospitals regarding their OPAT programs and structure. The fact that the authors were able to send a survey to all the hospitals is a major strength, as well as the high response rate from the institutions. As a practitioner in OPAT, it is very valuable to read about other program compositions and activities, to improve my own practice.
The survey was well written and went through several stages of vetting, before being sent out. This is also a strength of the manuscript.
As OPAT practice differs greats from country to country, the majority of comments are for additional background or clarity.
Consider moving up the description of who/where OPAT is performed from the discussion to the introduction. However, wherever the description remains, please expand upon it. Approach this as the reader is not familiar with the structure of healthcare in the Netherlands. Who/which pharmacy/hospital provides medication to the patients? Who infuses the medication? Does a nurse come daily to administer the medication? If yes, who pays for this? This is partially explained when discussing S-OPAT, but the non-S-OPAT population is not fully described. As a US-based reader, I was not clear on the healthcare set up.
In several locations, the authors mention that the OPAT team did not assume responsibility for the majority of the OPAT patients. What does the OPAT team do in this case, then? Are they setting up follow-up appointments? Coordinating medication and nursing visits? Reaching out for laboratory values?
Page 3, Results: Can the authors provide additional information why the 11 hospitals did not respond? How many of these were partial/incomplete responses and therefore not included? How many f/u attempts were made?
Table 1: Consider clarifying some terms that may be specific to the Netherlands. "Infectious disease specialist" - were these not all physicians? "Transfer of care specialist" - were these nurses? "Pharmacist assistant" - is this a "technician"?
Page 4, section 3.3: The authors provide information regarding how many OPAT patients per 100 hospital beds per year. Is there information regarding how many average active enrolled OPAT patients on any given day?
Table 2: Formatting issue: the results/p values don't line up with the characteristic rows. Some lines are missing numbers, other characteristics are written across 2-3 rows and each contain numbers
Table 2, "Presence of in- and exclusion criteria" Should this P value be "<0.001" rather than "0"?
Figure 1: Difficult to read when printed out as a pdf (versus viewing as a pdf on the computer). Please review before final printing. Additionally, perhaps rather than # of hospitals, this would be better represented as % of hospitals?
Figure 2: similar printing/readability issues as Figure 1. Please review before final version is published
Page 7, section 4. Please clarify the line, "...since the number of patients receiving OPAT annually differed between hospitals of similar size." Where was this information presented? Or if it was presented, I did not realize this and therefore this sentence isn't clear.
Page 8: The authors mention that OPAT programs would benefit from a staffing standard, which i agree with and support. An OPAT pharmacist to patient ratio has been suggested from US OPAT pharmacists: 1 pharmacist FTE for every 45-70 OPAT patients. Rivera et al. ASHE 2022, volume 2, issue 1, page e69. DOI: https://doi.org/10.1017/ash.2022.40
Supplement, line 19: Would recommend that the authors expand their description of the OPAT model. What is a home-based infusion? Where does the medication come from? Who administers it? Etc.
Overall I very much enjoyed reading and learning about the OPAT process in the Netherlands and appreciate the opportunity to review.
